# Measuring open defecation in India using survey questions: evidence from a randomised survey experiment

Sangita Vyas,[1,2] Nikhil Srivastav,[2,3] Divya Mary,[4] Neeta Goel,[5] Sujatha Srinivasan,[4] Ajaykumar Tannirkulam,[4] Radu Ban,[6] Dean Spears,[1,2,7] Diane Coffey[2,7,8]

For numbered affiliations see end of article.

**Correspondence to**
Sangita Vyas;
sangita.vyas@utexas.edu

## ABSTRACT

**Objectives** To investigate differences in reported open defecation between a question about latrine use or open defecation for every household member and a household-level question.

**Setting** Rural India is home to most of the world's open defecation. India's Demographic and Health Survey (DHS) 2015–2016 estimates that 54% of households in rural India defecate in the open. This measure is based on a question asking about the behaviour of all household members in one question. Yet, studies in rural India find substantial open defecation among individuals living in households with latrines, suggesting that household-level questions underestimate true open defecation.

**Participants** In 2018, we randomly assigned latrine-owning households in rural parts of four Indian states to receive one of two survey modules measuring sanitation behaviour. 1215 households were asked about latrine use or open defecation individually for every household member. 1216 households were asked the household-level question used in India's DHS: what type of facility do members of the household usually use?

**Results** We compare reported open defecation between households asked the individual-level questions and those asked the household-level question. Using two methods for comparing open defecation by question type, the individual-level question found 20–21 (95% CI 16 to 25 for both estimates) percentage points more open defecation than the household-level question, among all households, and 28–29 (95% CI 22 to 35 for both estimates) percentage points more open defecation among households that received assistance to construct their latrines.

**Conclusions** We provide the first evidence that individual-level questions find more open defecation than household-level questions. Because reducing open defecation in India is essential to meeting the Sustainable Development Goals, and exposure to open defecation has consequences for child mortality and development, it is essential to accurately monitor its progress.

**Trial registration number** Registry for International Development Impact Evaluations (5b55458ca54d1).

## INTRODUCTION

Rural India is home to more than half of the world's open defecation.[1] Because the persistence of open defecation threatens gains in child health, the Sustainable Development Goals (SDGs) call for its elimination

## Strengths and limitations of this study

► This is the first study that experimentally tests the difference in open defecation estimated from different survey questions.

► This study provides evidence that India's Demographic and Health Survey, which asks about the defecation behaviour of everyone in the household in one question, substantially underestimates open defecation in India.

► The study shows that measuring open defecation at the individual level is feasible in a large household survey, and finds more open defecation than household-level questions.

► Monitoring open defecation in India is important for understanding progress towards the Sustainable Development Goals.

► Because the households participating in this study are not representative of the rural parts of the states they are in, or rural India, the estimates presented here should not be considered as estimates of open defecation for any of the states, or the country as a whole.

by 2030. Progress towards eliminating open defecation in rural India will be essential to meeting this goal. India's most recent Demographic and Health Survey (DHS), conducted between January 2015 and December 2016, estimates that 54% of households in rural India defecated in the open, down from 75% in the 2005 to 2006 DHS.[2 3] This measure is based on a household-level question that asks about the behaviour of everyone in the household in the same question.

Recent evidence from studies carried out in India suggests, however, that it is common for individuals living in households with latrines to nevertheless defecate in the open. In rural parts of five north Indian states, Coffey *et al* found that 21% of individuals defecated in the open, despite owning a latrine.[4] In rural Tamil Nadu, Yogananth and Bhatnagar report that 54% of respondents defecated in the open despite having a household latrine.[5]

In Odisha, Barnard *et al* found that less than half of members of households with latrines reported using their latrines at all times.[6]

Experimental studies of sanitation interventions have found similar results. Clasen *et al* report on a sanitation intervention in Odisha and note open defecation among individuals living in households with latrines as a reason for not observing impacts on child health outcomes.[7] Patil *et al* conducted a sanitation intervention in Madhya Pradesh and experienced a similar problem: modest increases in latrine coverage, and even more modest reductions in open defecation.[8] These findings suggest that open defecation among latrine-owning households is substantial. Since it is probable that latrine use is the socially desirable response to questions on sanitation behaviour, measures based on household-level questions, such as those from the DHS, will likely underestimate true open defecation in rural India, particularly among households with latrines.

Because open defecation is an individual behaviour, an individual-level survey question may be able to more accurately measure it compared with a household-level question, particularly among households with latrines. We designed this study to experimentally test this hypothesis in rural India. We aimed to investigate whether a balanced question about latrine use or open defecation for every member of a household finds different levels of open defecation compared with a household-level question.

This is the first study to experimentally vary survey methodology to improve on the measurement of open defecation currently being used. Jenkins *et al* study sanitation survey methods, and report on an index they develop for quantifying household excreta disposal.[9] Their study focuses on developing and piloting a new tool rather than comparing different measures. Sinha *et al* compare answers to survey questions on latrine use behaviour to measures of actual behaviour generated from passive latrine use monitors that were set up in the latrines of respondents, and find poor to moderate agreement between the two measures.[10] Our study contributes to this literature by comparing estimates of open defecation obtained from questions that can be administered in a large household survey, and highlighting potential sources of error in open defecation measurement.

Four years ago, the Government of India launched the Swachh Bharat Mission (SBM), a national sanitation campaign, which aims to eliminate open defecation in India by 2019. Many latrines have been constructed in rural India as a result of this campaign. Yet, the effect the SBM has had on reducing open defecation is still unknown. Because large reductions in open defecation in India are essential to meeting the SDGs, and because exposure to open defecation has serious consequences for child mortality, health and human capital development, it is essential to monitor its progress as accurately as possible.

## METHODS
The study is registered in the Registry for International Development Impact Evaluations.

### Sample: mostly latrine-owning households in rural parts of four states
This study uses as its sampling frame the study areas of 3ie's Promoting Latrine Use in Rural India Thematic Window. This window has funded four independent research teams to conduct randomised control trials of distinct behavioural campaigns to promote the use of pit latrines in rural parts of Bihar, Gujarat, Karnata and Odisha. The study areas are spread across India, representing different contexts and varying levels of rural open defecation.

Because these trials focus on behavioural strategies rather than latrine construction, they are being carried out in villages that had high levels of coverage of pit latrines at baseline, relative to other rural parts of the same states. The households that comprise the sampling frame for this study are those that were identified as having a functional latrine in a census conducted by the research teams in the villages in which they were working. In all states except for Odisha, only households that had been excluded from the research teams' samples could be selected for this study. In Odisha, the sample selected for this study overlaps with the research teams' sample. We aimed to survey households that own latrines because we expect that an important source of misreporting of open defecation comes from individuals who do not use the functional latrines that their households own, and that other members of their households use.

Figure 1 describes the sample selection. The villages visited in each state were randomly selected from the full set of villages included in the 3ie research teams' studies. The full set of villages were selected by the research teams in collaboration with the implementation agencies they were working with. The research team led by Oxford Policy Management worked with World Vision in Bihar, the team led by London School of Hygiene and Tropical Medicine worked with Coastal Salinity Prevention Cell in Gujarat, Eawag worked with Wateraid in Karnataka, and Emory University worked with the Rural Welfare Institute in Odisha. Data for our study were collected in 22–25 villages in each of the four study areas. In most areas, we sampled more villages than we actually visited in order to facilitate coordination with the research teams. Ninety-five villages were visited in total.

Up to 40 households in each village were randomly assigned to receive the household or individual questions. In some villages, fewer than 40 households were assigned because fewer than 40 households met the eligibility criteria. The survey team visited as many assigned households as it could in these villages, given time constraints, and availability of household members. On average, the survey team interviewed 25 households per village. Data collection took place between March and July 2018.

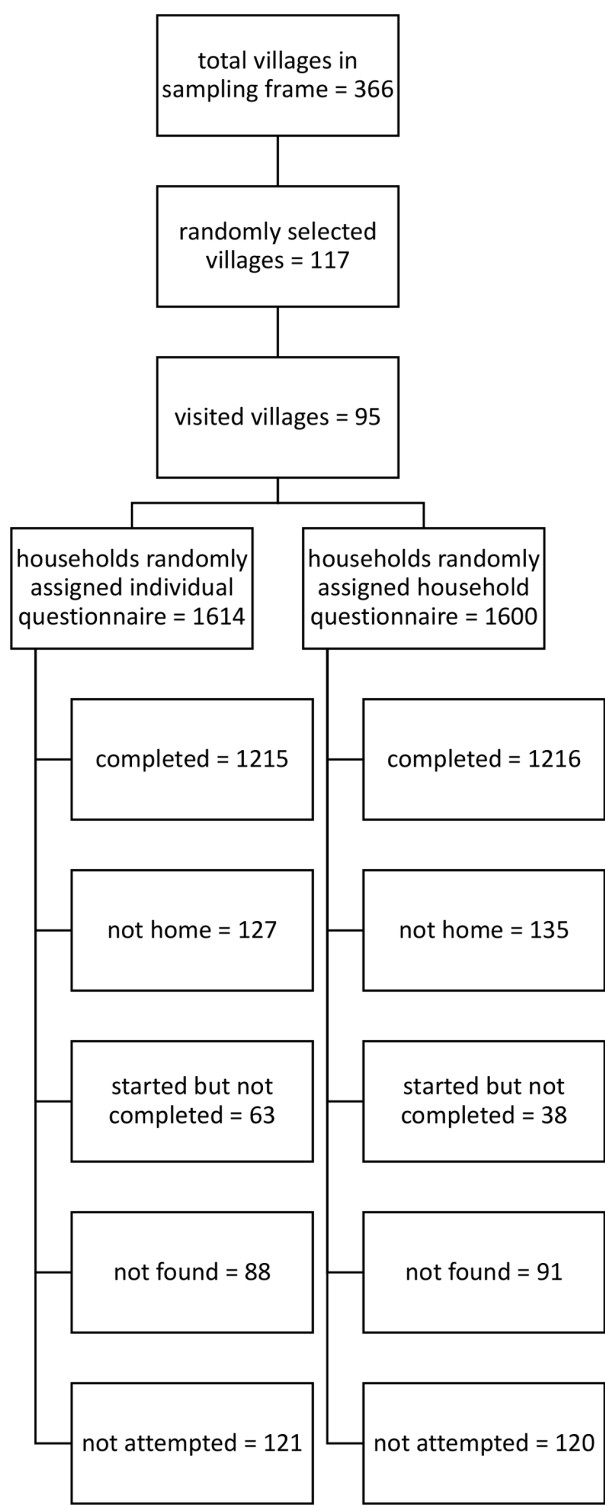

**Figure 1** Sample selection. Households that were started but not completed refers to households that refused at the beginning or part of the way through and households in which a suitable respondent was not available.

In each state, data collection took place after the 3ie research teams had conducted their censuses and baselines, but before they had started their interventions. Since in all states, the households visited in this study were also visited at the time of the census, response bias may be a concern. This would not, however, impact the internal validity of this study since randomisation generates equal response bias, in expectation, across treatment arms.

## Randomisation and masking: random variation in latrine use questions at the household level

We randomly assigned the type of latrine use question administered in the survey at the household level. Roughly half of the households were assigned individual-level questions on latrine use. The other half were assigned a household-level question. One of the authors who was not involved in data collection carried out the randomisation using a random number generator in Stata. Because of the nature of the study, it was not possible to blind the respondents or surveyors to the type of survey question administered in the survey. However, in the interest of data quality, respondents were not explicitly told that the primary purpose of the survey, which took ~25 min to complete, was to measure open defecation. Additionally, surveyors did not know which survey question had been assigned to a household until starting the survey with the household. This was facilitated through SurveyCTO, the mobile data collection platform used in the study, which was programmed to store the randomisation assignment for each household ID prior to the commencement of data collection. In the field, surveyors were only given a list of households to interview. When a surveyor had correctly identified a household and was ready to start the survey, she would enter the household ID into SurveyCTO, and SurveyCTO would automatically start the questionnaire type assigned to the household.

The individual-level questions asked for every household member age 5 or older whether the individual defecated in the open or used the latrine. The preface to this series of questions was: 'I have seen that some people defecate in the open, and some people use the latrine. Now I want to ask about where you and your family members defecate'. Then, the surveyor asked the following question for each individual in the household, and coded the answer in a household roster: 'The last time [*name of household member*] defecated, did [*name of household member*] defecate in the open or use the latrine?'. The answer options included latrine, open and somewhere else. Surveyors used the last option, which meant that the household member defecated in a bedpan, cloth or other place, in <0.5% of cases. Because the priming statement and the behaviour question include both open defecation and latrine use, they are balanced between the two different behaviours and could reduce social desirability bias. The surveyor asked household members who were participating in the interview directly about their behaviour, and asked the main respondent, in most cases an adult female member of the household, to report on the behaviour of their family members who were not participating in the interview.

The rest of the households were assigned the household-level question used in India's DHS: 'What kind of toilet facility do members of your household usually use?'.[11] The answer codes were also the same as those

used in India's DHS: flush to piped sewer system, flush to septic tank, flush to pit latrine, flush to somewhere else, flush to don't know where, ventilated improved pit or biogas latrine, pit latrine with slab, pit latrine without slab or open pit, twin pit or composting toilet, dry toilet and no facility or uses open space or field. We also included an individual-level question on mobile ownership or preferring vegetarian food versus non-vegetarian food in the surveys that asked the household-level question so that both types of surveys would take approximately the same amount of time to complete.

There are three main factors that differ between the two types of latrine use survey modules: the level of aggregation, the reference period and the presence of a priming statement. Therefore, the differences in reported open defecation that we observe reflect the fact that the two sets of questions vary on all of these factors combined.

### Statistical analyses

The primary outcome of interest is reported open defecation. For the household-level questions, we created a dummy variable that is equal to one if the response was 'no facility or uses open space or field', and zero otherwise. The unit of observation for households assigned the individual-level question is the individual, while the unit of observation for those assigned the household-level question is the household. Therefore, in order to directly compare and test the significance of differences in reported open defecation between the two question types, we construct estimates that use the same unit of observation. We impute individual-level open defecation from responses to the household-level questions, and household-level open defecation rates from responses to the individual-level questions. To construct individual-level open defecation using the household-level questions, we assign the answer from the household question to each individual in the household. Similarly, to construct household-level open defecation using the individual-level questions, we average the responses among individuals in the household, and assign this average as the household value. Our main analysis tests differences in measured open defecation by question type. We show pooled results, as well as results by study area.

We also conduct subgroup analyses. These analyses investigate differences in the same primary outcome measure, reported open defecation, but look at differences by question type among different subgroups. First, we investigate whether the difference in reported open defecation by question type depends on whether the latrine was constructed privately, or with assistance from the government or a non-governmental organization (NGO). In practice, assistance to construct latrines often comes from the government, but sometimes NGOs get involved in facilitating the implementation of the government programme. As part of the SBM, the Government of India assists rural households to construct latrines either by providing financial assistance directly to households so they can construct their own latrines or by local

government officials constructing latrines for households. In the discussion that follows, we will describe a household as having 'received help' if it received financial assistance or a partially or completely constructed latrine from the government or an NGO.

The Indian government promotes and constructs latrines with pits that are ~60 ft³.[12] However, many rural Indians aspire to construct latrines with pits that are much larger, so that they can avoid emptying the pit, a task that is associated with ritual pollution.[13–16] Compared with latrines constructed privately, those constructed with government help are less likely to be used due to concerns over purity and pit emptying. Since a large fraction of rural households are likely to receive latrines with help from the government as a result of the SBM, it is important to explore how much open defecation different types of latrine use questions measure, based on having received help to construct the latrine.

The second subgroup analysis investigates whether the difference in measured open defecation between the two question types is statistically different for males compared with females. Sex differences are an important aspect to explore because observational studies have found consistently higher open defecation among latrine owners for males compared with females.[4] This observation could reflect greater demand for latrine use among females due to, for instance, greater psychosocial stress experienced when defecating in the open,[17] or it could be because of cultural norms that keep females in their reproductive years inside the home.

Means and differences in reported open defecation, by question type, are calculated using ordinary least squares regression with cluster robust SEs, clustered by village. Statistical analyses were conducted using Stata V.11.

### Public involvement

The individual-level questions used in this study are the product of a deliberative process between the authors and research teams from Oxford Policy Management, London School of Hygiene and Tropical Medicine, Indian Institute of Public Health Gandhinagar, Eawag and Emory University. The findings of this study contribute to our understanding of the scale of an important public health problem.

### RESULTS

Table 1 shows summary statistics for households assigned the two types of latrine use questions. The total sample consisted of 2431 households, which were approximately equally divided across question type in each of the study areas. There were no significant differences on measures relevant for latrine use between households assigned different types of latrine use questions. Households in both groups had approximately the same number of household members, fraction female, fraction Hindu, educational attainment of the household head, and asset ownership of 13 assets, including mobile phone,

**Table 1** Randomisation balance: no significant differences in observed means between households assigned different latrine use questions

|  | Individual | Household | Difference |
|---|---|---|---|
|  | (1) | (2) | (3) |
| No of households | 1215 | 1216 | −1 |
| No of households by project |  |  |  |
| World Vision (in Bihar) | 316 | 313 | 3 |
| Coastal Salinity Prevention Cell (in Gujarat) | 309 | 319 | −10 |
| Wateraid (in Karnataka) | 297 | 296 | 1 |
| Rural Welfare Institute (in Odisha) | 293 | 288 | 5 |
| Household members | 5.685 (0.0950) | 5.604 (0.101) | 0.0803 (−0.117 to 0.278) |
| Female | 0.491 (0.00485) | 0.489 (0.00490) | 0.00190 (−0.0115 to 0.0153) |
| Hindu | 0.967 (0.00882) | 0.960 (0.0102) | 0.00737 (−0.00383 to 0.0186) |
| Household head completed at least 8 years of schooling | 0.288 (0.0192) | 0.319 (0.0199) | −0.0310 (−0.0675 to 0.00552) |
| Count of assets (max 13) | 8.202 (0.111) | 8.234 (0.117) | −0.0327 (−0.196 to 0.130) |
| Has latrine | 0.943 (0.00795) | 0.946 (0.00781) | −0.00251 (−0.0215 to 0.0165) |
| Got help from government or NGO to build toilet (given has toilet) | 0.625 (0.0304) | 0.656 (0.0274) | −0.0309 (−0.0692 to 0.00747) |
| Pit size (cubic feet, given has toilet) | 179.0 (14.33) | 180.7 (14.27) | 1.666 (−25.94 to 22.61) |
| Toilet looks used (given has toilet) | 0.805 (0.0234) | 0.794 (0.0233) | 0.0103 (−0.0243 to 0.0450) |

Cluster robust SEs, clustered by village, next to means in columns 1 and 2. 95% CI next to differences in column 3. *P<0.05, **p<0.01. None of the differences are significant at the 0.05 or 0.01 level, which is why there are no * or ** in the table.

electricity, radio, television, fan, mosquito net, bicycle, motorcycle, car, chair, gas stove, pressure cooker and shoes for everyone in the family. It is important for the validity of the results that the sample is balanced on religious composition, since studies have documented an association between household religion and latrine use.[18 19]

As the study design intended, most households in both groups had a latrine. Conditional on having a latrine, 64% of households had gotten help from the government or an NGO to build the latrine, and the average pit size was 180 ft$^3$. It is important that the sample is balanced on these two characteristics because, compared with latrines constructed privately, those constructed with government help are less likely to be used because of concerns over purity and pit emptying.[13–16] Finally, among households with latrines, ~80% of them appeared to the surveyor to be in use on observation.

Figure 2 presents the main results of the study; it shows means and 95% CIs from the individual-level (shown as dashed red bars) and the household-level (shown as solid blue bars) questions, for the full dataset and for different subsamples. Observations are individuals for the individual-level estimates, and households for the household-level estimates. In the full sample, and in all subsamples analysed in this figure, the individual questions find more open defecation. Moreover, the size of the difference in reported open defecation between question types is large and consistent. The first set of estimates shown in the figure uses the full sample. The second, third, fourth

and fifth sets of estimates break the sample up by project area. The sixth set of estimates uses only households with latrines. No matter how the data are broken up, the individual-level, balanced latrine use questions find significantly higher rates of open defecation than the household-level question.

Table 2 shows actual and imputed open defecation rates, measured at the individual and household levels. Columns 1 and 3 are the estimates shown in figure 2. Columns 2 and 4 show imputed open defecation rates, at the individual and household level, respectively. Imputed values are calculated based on the method described in the Statistical analyses section. Column 5 shows the difference in the measured rate of open defecation between the individual and household questions when observations are individuals, and column 6 shows the same difference when observations are households.

In the full sample, the individual-level, balanced questions find 21 (95% CI 16 to 25) percentage points more open defecation than the household-level question when observations are individuals, and 20 (95% CI 16 to 25) percentage points more open defecation when observations are households. Notably, the individual-level questions measure consistently higher levels of open defecation in the full sample and in all subsamples, irrespective of how the difference is calculated. All differences are significant at the 1% level.

The seventh set of estimates shown in figure 2 show reported open defecation from individual-level and household-level questions among households that

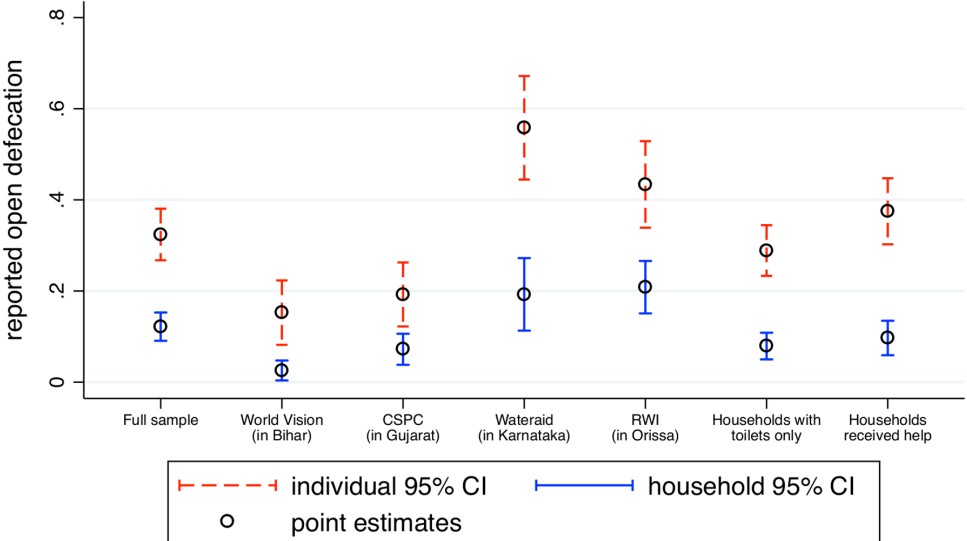

**Figure 2** Individual-level, balanced latrine use questions find significantly higher rates of open defecation than the household-level question. Figure shows means and CIs. Red, dashed lines indicate responses to individual-level, balanced latrine use questions, and blue solid lines indicate responses to the household-level question. Unit of observation is individuals for individual-level questions and households for household-level questions. CIs are computed using SEs clustered by village.

received help to construct their latrines. Comparing these estimates to the sixth set of estimates in the figure, which include all households with latrines, suggests that the household-level question underestimates open defecation by more among those that received help to construct their latrines, compared with those that did not. The seventh row in table 2 shows that, using different methods for computing differences, individual-level questions measure 28–29 (95% CI 22 to 35 for both estimates) percentage points more open defecation than the household-level question, among those that received help to construct their latrines, compared with those that did not.

The first two columns in table 3 test whether the difference in measured open defecation between the two question types is statistically different among households that received help to construct their latrines compared with households that did not receive help to construct their latrines. The first column in table 3 uses individuals as observations, and the second column uses households. The coefficients in the third row represent the difference-in-differences estimate.

The difference in measured open defecation between the two question types is 18 (95% CI 11 to 25) to 19 (95% CI 11 to 26) percentage points larger among households that received help to construct their latrines, compared with households that did not, depending on the method used to compute the difference. Among households that did not receive help to construct their latrines, the individual-level questions find 10 (95% CI 5 to 15) percentage points more open defecation than the household-level question.

The last column in table 3 investigates whether the difference in measured open defecation between the two question types is statistically different among males compared with females. Using the full sample, individual questions find 4 (95% CI 2 to 7) percentage points more open defecation than the household question among males compared with females. Breaking the sample up by sex, the individual questions find 23 (95% CI 18 to 28) percentage points more open defecation than the household question among males, and 19 (95% CI 14 to 24) percentage points more among females. The household question underestimates open defecation by more among males compared with females.

## DISCUSSION

Our findings show that in our sample, individual-level, balanced questions find 20–21 (95% CIs 16 to 25 for both estimates) percentage points more open defecation than the household-level question. This is both a statistically significant and practically important difference. This study presents compelling evidence that India's DHS, which provides the most recent nationally representative estimates of open defecation for rural India, and other surveys that ask household-level questions, greatly underestimate open defecation among households with latrines.

We also found that the difference in reported open defecation between the two question types is significantly greater for households that received help to construct their latrines compared with households that did not. Among households that received help to construct their latrines, the individual-level questions find 28–29 (95% CIs 22 to 35 for both estimates) percentage points more open defecation than the household-level question. This suggests that as more and more households receive government assistance for a latrine through the SBM, household-level questions will become even less accurate at estimating open defecation.

**Table 2** Individual-level, balanced latrine use questions find significantly higher rates of open defecation than household level questions

| Unit of observation: | Individuals | | Households | | | |
|---|---|---|---|---|---|---|
| Question type | Individual | Household (imputed) | Household | Individual (imputed) | Difference (1)–(2) | Difference (4)–(3) |
| | (1) | (2) | (3) | (4) | (5) | (6) |
| **1. Full sample** | | | | | | |
| Estimates | 0.324 (0.0288) | 0.115 (0.0150) | 0.122 (0.0158) | 0.326 (0.0289) | 0.209** (0.163 to 0.254) | 0.204** (0.160 to 0.248) |
| n (individuals or households) | 13070 | 13070 | 2431 | 2431 | | |
| **2. World Vision sample (in Bihar)** | | | | | | |
| Estimates | 0.153 (0.0361) | 0.0184 (0.00796) | 0.0256 (0.0112) | 0.154 (0.0321) | 0.134** (0.0659 to 0.202) | 0.128** (0.0719 to 0.184) |
| n (individuals or households) | 3675 | 3675 | 629 | 629 | | |
| **3. Coastal Salinity Prevention Cell sample (in Gujarat)** | | | | | | |
| Estimates | 0.192 (0.0358) | 0.0848 (0.0212) | 0.0721 (0.0174) | 0.169 (0.0299) | 0.108** (0.0355 to 0.180) | 0.0973** (0.0417 to 0.153) |
| n (individuals or households) | 3340 | 3340 | 628 | 628 | | |
| **4. Wateraid sample (in Karnataka)** | | | | | | |
| Estimates | 0.558 (0.0580) | 0.184 (0.0365) | 0.193 (0.0406) | 0.550 (0.0575) | 0.374** (0.262 to 0.486) | 0.358** (0.247 to 0.468) |
| n (individuals or households) | 3112 | 3112 | 593 | 593 | | |
| **5. Rural Welfare Institute sample (in Odisha)** | | | | | | |
| Estimates | 0.434 (0.0484) | 0.204 (0.0284) | 0.208 (0.0294) | 0.450 (0.0463) | 0.230** (0.150 to 0.309) | 0.242** (0.170 to 0.313) |
| n (individuals or households) | 2943 | 2943 | 581 | 581 | | |
| **6. Households with toilets only** | | | | | | |
| Estimates | 0.289 (0.0284) | 0.0734 (0.0137) | 0.0791 (0.0150) | 0.291 (0.0287) | 0.215** (0.166 to 0.265) | 0.211** (0.162 to 0.261) |
| n (individuals or households) | 12366 | 12366 | 2296 | 2296 | | |
| **7. Households that received help** | | | | | | |
| Estimates | 0.375 (0.0370) | 0.0895 (0.0181) | 0.0968 (0.0193) | 0.378 (0.0368) | 0.285** (0.220 to 0.351) | 0.281** (0.217 to 0.346) |
| n (individuals or households) | 7958 | 7958 | 1470 | 1470 | | |

Cluster robust SEs, clustered by village, under means in columns 1 through 4. 95% CI under differences in columns 5 and 6. *P<0.05, **p<0.01. Household imputed refers to estimates of individual open defecation imputed from answers to the household-level question. Each individual in the household is given the same answer as the household-level answer. Individual imputed refers to estimates of household open defecation imputed from answers to the individual level questions. The household estimates are constructed by averaging open defecation among individuals in the household. There are no differences that are significant at the .05 level, all are significant at the .01 level. That is why * is not used in the table.

**Table 3** Subgroup analyses

| Unit of observation | Individuals | Households | Individuals |
|---|---|---|---|
| | (1) | (2) | (3) |
| Individual-level question | 0.100** (0.0548 to 0.146) | 0.0992** (0.0566 to 0.142) | 0.229** (0.182 to 0.275) |
| Received help for construction | 0.0470* (0.00962 to 0.0844) | 0.0514** (0.0134 to 0.0893) | |
| Individual-level question × received help | 0.185** (0.113 to 0.257) | 0.182** (0.114 to 0.251) | |
| Female | | | 0.00134 (−0.0118 to 0.0144) |
| Individual-level × female | | | −0.0409** (−0.0646 to −0.0172) |
| Constant | 0.0425** (0.0180 to 0.0670) | 0.0455** (0.0201 to 0.0708) | 0.115** (0.0846 to 0.145) |
| n (individuals or households) | 12366 | 2296 | 13070 |

95% CI next to coefficients, calculated using cluster robust SEs, clustered by village. *P<0.05, **p<0.01.

The larger difference in measured open defecation between the two question types among households that received help compared with those that did not is likely arising from higher rates of open defecation among households that received help to construct their latrines. There are several reasons that could explain why households receiving assistance may be less likely to use their latrines. First, these households are likely to have lower demand for latrine use, compared with households that built latrines on their own. Second, households that received help have latrines with pits that are on average 150 ft$^3$ smaller than the pits of latrines in households that did not receive help. Because of concerns over ritual purity, rural Indians are less likely to use latrines with pits that need to be emptied manually every few years, like the latrines that are promoted and constructed by the government.[13–16] Whether only one, both or other factors are leading to more open defecation among households that received help, the individual-level questions are better able to capture this open defecation than the household-level question.

We also find a statistically significant difference in reported open defecation between the two question types for males compared with females. The difference between the individual-level, balanced questions and the household-level question is 4 (95% CI 2 to 7) percentage points more for males compared with females. This supports evidence that, conditional on latrine ownership, males are more likely to defecate in the open compared with females. Individual-level questions understate the difference in open defecation between the two sexes by less than the household-level questions. Compared with the difference in reported open defecation by receiving help to construct the latrine, differences by sex are not as large.

Measuring open defecation at the individual level is feasible. Our survey team's experience suggests that adding the balanced, individual-level questions on use to a survey that already contains a household roster increases survey time by about 2 min, on average. Of course, the amount of time required to ask the individual-level questions depends on the number of individuals in the household.

The household-level question asked in the DHS also collects information on the types of latrines that households own, data that are still of great interest to researchers and practitioners. Therefore, individual questions on use, combined with a separate question on the types of latrines that households own, would satisfy both goals: evaluating latrine infrastructure, and measuring open defecation as accurately as possible.

A limitation of our study is that the samples from these project areas are not representative of the rural parts of the states they are in, nor are they collectively representative of rural India. The households in this study are much more likely to have a latrine than the average rural Indian household, and therefore, the individuals in this study are more likely to use a latrine. For this reason, the estimates presented here should not be considered as estimates of open defecation for any of the states, or the country as a whole. Rather, they show a large and significant difference in reported open defecation based on the type of question asked.

Measuring open defecation at the individual-level in a large household survey is doable and will provide a more accurate estimate of open defecation in rural India. Since reducing open defecation in India is important for meeting the SDGs, and since open defecation is an important factor contributing to poor health among children in India, it is important to measure its progress as accurately as possible.

**Author affiliations**
[1]Economics and Population Research Center, University of Texas at Austin, Austin, Texas, USA
[2]r.i.c.e, India
[3]LBJ School of Public Affairs, University of Texas at Austin, Austin, Texas, USA
[4]IFMR LEAD, Institute for Financial Management and Research, Chennai, India
[5]International Initiative for Impact Evaluation, New Delhi, India
[6]Evidence and Measurement, WSH Program, Bill and Melinda Gates Foundation, Seattle, Washington, USA
[7]Indian Statistical Institute, New Delhi, India
[8]Sociology and Population Research Center, University of Texas at Austin, Austin, Texas, USA

**Acknowledgements** We are very grateful for the collaboration and contributions of the 3ie research teams in formulating the individual-level questions. The questions are the product of a deliberative process between the authors and the research

teams. We are also grateful for the research teams' cooperation on this study. In particular, we thank researchers from Oxford Policy Management, London School of Hygiene and Tropical Medicine, Indian Institute of Public Health Gandhinagar, Eawag, and Emory University, and the implementing agencies with which they are working, including World Vision, Coastal Salinity Prevention Cell, Wateraid, and Rural Welfare Institute, for sharing their census data with us, coordinating field work and supporting this endeavor. This study would not have been possible without their collaboration and assistance.

**Contributors** SV, NS, DS and DC contributed to the study design. SV, NS, DC and DS designed the survey instruments. NS, DM, SS and AT oversaw data collection. SV, NS and NG coordinated between the 3ie research teams and the survey team for this study. SV, RB, DC and DS contributed to analysis. All authors contributed to drafting the report.

**Funding** International Initiative for Impact Evaluation, Bill & Melinda Gates Foundation. RB, who is employed on the WSH programme of BMGF, contributed to the analysis of the results. NG contributed to coordinating data collection. Both contributed inputs to drafting the report.

**Competing interests** None declared.

**Patient consent for publication** Not required.

**Ethics approval** The study received ethical approval for research involving human subjects from the Institute for Financial Management and Research's Institutional Review Board in India, Approval # IRB00007107.

**Provenance and peer review** Not commissioned; externally peer reviewed.

**Data availability statement**
Data are publicly available here: https://riceinstitute.org/data/measuring-open-defecation-in-india-using-survey-questions-a-randomized-survey-experiment/

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
