## [Reviewer comments · BMJ Open]

ARTICLE DETAILS

TITLE (PROVISIONAL)	Measuring open defecation in India using survey questions: Evidence from a randomized survey experiment
AUTHORS	Vyas, Sangita; Srivastav, Nikhil; Mary, Divya; Goel, Neeta; Srinivasan, Sujatha; Tannirkulam, Ajaykumar; Ban, Radu; Spears, D; Coffey, Diane

VERSION 1 – REVIEW

REVIEWER	Dr. Mohammad Rashid Indian Institute of Technology Kharagpur
REVIEW RETURNED	18-Mar-2019

GENERAL COMMENTS	Dear Editor The paper is of very good quality and very well written. To my knowledge, this is the first paper in which open defecation is measured at individual level and compared with the household level responses. There are no queries from my side and recommend to accept the paper. Thank you
--

REVIEWER	David Blanco Universitat Politècnica de Catalunya
REVIEW RETURNED	

GENERAL COMMENTS	This report shows the results of an evaluation of the consistency between the CONSORT checklist you submitted and the information that was reported in the manuscript. The examples or cites included in the report were extracted from the CONSORT E&E Document (https://www.bmj.com/content/340/bmj.c869). Please, make the following revisions: For CONSORT Item 6a ("Completely defined pre-specified primary and secondary outcome measures, including how and when they were assessed"), please explicitly specify in the text whether the outcomes mentioned in pg. 7-8 are primary, i.e. the outcome of greatest importance to relevant stakeholders, or secondary. For example, on the second paragraph of pg. 8 I would suggest to start saying: "We considered two secondary outcomes arising from a subgroup analysis. The first secondary outcome was the difference in open defecation between participants whose latrines were constructed privately and those that had assistance from the government or an NGO [...]". Please follow this type of structure for the other outcomes. Presenting the study outcomes transparently makes the study results more straightforward to understand.
---

	 • For CONSORT Item 9a (“Mechanism used to implement the random allocation sequence, describing any steps taken to conceal the sequence until interventions were assigned”), please explain what strategy was followed to implement the random allocation, that is, how surveyors were told which survey question had been assigned to a certain household and how you ensured that this information was concealed until the start of the survey. As CONSORT E&E document states, please bear in mind that “allocation concealment should not be confused with blinding. Allocation concealment seeks to prevent selection bias, protects the assignment sequence until allocation, and can always be successfully implemented. In contrast, blinding seeks to prevent performance and ascertainment bias, protects the sequence after allocation, and cannot always be implemented”. • For CONSORT Item 13b (“For each group, losses and exclusions after randomisation, together with reasons”), please include in the flow diagram the number of lost to follow-up participants for each group and provide the reasons why this happened. It would be of great importance to make clear how often and why some households were assigned to be interviewed but did not complete these interviews.  o An example of proper reporting of losses and exclusions after randomisation can be found in Fig. 3 of the CONSORT E&E document (http://www.consort-statement.org/Media/Default/Downloads/CONSORT%202010%20Explanation%20and%20Elaboration%20Document-BMJ.pdf)
--	---

REVIEWER	Robert Ntozini Zvitambo Institute for Maternal and Child Health Research Harare, Zimbabwe
REVIEW RETURNED	02-Apr-2019

GENERAL COMMENTS	Title: Measuring open defecation in India using survey questions: Evidence from a randomized survey experiment This review is limited to the statistical methods used in the paper. The paper adds to the methods of assessing open defecation in rural settings which is important in meeting the SDGs. The researchers randomized latrine owning households to either complete the usual India DHS survey questionnaire on household latrine which is asked at household level or a questionnaire administered to each individual member of the household to solicit their toilet behaviors. Overall, the study was well conducted and clearly described. Strengths of the study design were the randomization, making the length of the surveys similar and keeping the surveyors blinded to the actual survey questions until they started the survey with the household. A potential weakness of the study design was the selection of villages and individuals within the villages; the authors acknowledge that they selected more villages than they could assess in order to “facilitate coordination with the research teams”, and that the survey teams visited as many assigned households as they could in these villages, given “time constraints, and availability of household members”. Easy to reach villages and households where members were easily available may have different toilet behaviors than those not reached which is a potential selection bias. The authors however acknowledge that the estimates obtained are limited to the sample and do not represent the population they
--

	were selected from, which makes the bias less important. The analysis approach of comparing households that responded to individual level questions to household that responded to individual level questions by imputing individual responses from household responses and aggregating individual responses into household responses is a valid approach given the data. I have the following comments for the authors to consider in further revisions. Major comments: -In the method section the authors only state that they clustered standard errors at the village level, however they do not state which method they employed to compute the mean differences. I recommend that authors state the regression methodology that used and whether they computed robust standard errors.-In the abstract, without reference to the result tables, the main results are potentially confusing because the comparison groups are not readily discernible. The authors could consider rephrasing the results to include the absolute proportions found by each method as well as the differences. For example the results could be stated as "Open defecation levels were 32 to 33% in households that responded to individual questions compared to 12% in households which responded to household questions, a difference of 20 to 21% 95% CI: 16 to 25 (for both estimates)" Minor Comments: -The presentation of findings could be improved, the authors tended to explain each table and figure presented in the results rather that state what the findings are.-It seems that figure 2 and table 2 both present the same results. I recommend that the authors choose which method best summarizes their finding.
--	--

VERSION 1 – AUTHOR RESPONSE

Reviewer: 1

1. The paper is of very good quality and very well written. To my knowledge, this is the first paper in which open defecation is measured at individual level and compared with the household level responses. There are no queries from my side and recommend to accept the paper.

Response: Thank you for reading the paper!

Reviewer: 2

1. For CONSORT Item 6a ("Completely defined pre-specified primary and secondary outcome measures, including how and when they were assessed"), please explicitly specify in the text whether the outcomes mentioned in pg. 7-8 are primary, i.e. the outcome of greatest importance to relevant stakeholders, or secondary. For example, on the second paragraph of pg. 8 I would suggest to start saying: "We considered two secondary outcomes arising from a subgroup analysis. The first secondary outcome was the difference in open defecation between participants whose latrines were constructed privately and those that had assistance from the government or an NGO [...]". Please follow this type of structure for the other outcomes. Presenting the study outcomes transparently makes the study results more straightforward to understand.

Response: Thank you for raising this issue. We have modified the text on pages 7 and 8 to clarify the discussion of the analyses. In particular, this paper only investigates one outcome measure, reported open defecation, which is the primary outcome measure. The main analysis investigates reported open defecation by question type, pooled and by study area. In supplementary analyses, we investigate differences in reported open defecation by question type, among different subgroups of the sample.

2. For CONSORT Item 9a ("Mechanism used to implement the random allocation sequence, describing any steps taken to conceal the sequence until interventions were assigned"), please explain what strategy was followed to implement the random allocation sequence, that is, how surveyors were told which survey question had been assigned to a certain household and how you ensured that this information was concealed until the start of the survey. As CONSORT E&E document states, please bear in mind that "allocation concealment should not be confused with blinding. Allocation concealment seeks to prevent selection bias, protects the assignment sequence until allocation, and can always be successfully implemented. In contrast, blinding seeks to prevent performance and ascertainment bias, protects the sequence after allocation, and cannot always be implemented".

Response: We have added more description, on page 6, of the process used for ensuring that surveyors only became aware of the randomization assignment when the survey started. This was facilitated through SurveyCTO, the mobile data collection platform used in the study, which was programmed, prior to the commencement of data collection, to store the randomization assignment for each household ID. In the field, surveyors were only given a list of households to interview. When a surveyor had correctly identified a household and was ready to start the survey, she would enter the household ID into SurveyCTO, and SurveyCTO would automatically start the questionnaire type assigned to the household.

3. For CONSORT Item 13b ("For each group, losses and exclusions after randomisation, together with reasons"), please include in the flow diagram the number of lost to follow-up participants for each

group and provide the reasons why this happened. It would be of great importance to make clear how often and why some households were assigned to be interviewed but did not complete these interviews.

o An example of adequate reporting of losses and exclusions after randomisation can be found in Fig. 3 of the CONSORT E&E document (<http://www.consort-statement.org/Media/Default/Downloads/CONSORT%202010%20Explanation%20and%20Elaboration%20Document-BMJ.pdf>)

Response: Thanks for this comment. Please see the revised Figure 1, which now shows categories for why households that were randomized did not complete an interview.

Reviewer: 3

1. In the method section the authors only state that they clustered standard errors at the village level, however they do not state which method they employed to compute the mean differences. I recommend that authors state the regression methodology that used and whether they computed robust standard errors.

Response: Thank you for raising this issue. We have modified the discussion of the statistical analyses on page 9 to clarify how we conduct our analysis. In particular, means and differences in reported open defecation, by question type, are calculated using ordinary least squares regression with cluster robust standard errors, clustered by village. We have also modified the notes under each of the tables to clarify how errors are structured.

2. In the abstract, without reference to the result tables, the main results are potentially confusing because the comparison groups are not readily discernible. The authors could consider rephrasing the results to include the absolute proportions found by each method as well as the differences. For example the results could be stated as “Open defecation levels were 32 to 33% in households that responded to individual questions compared to 12% in households which responded to household questions, a difference of 20 to 21% 95% CI: 16 to 25 (for both estimates)”

Response: Thanks so much for this comment. We have modified the abstract to clarify the two groups of interest for comparing reported open defecation. The results section in the abstract now reads as follows: “We compare reported open defecation between households asked the individual-level questions and those asked the household-level question. Using two methods for comparing open defecation by question type, the individual-level question found 20 to 21 (95% CI 16 to 25 for both estimates) percentage points more open defecation than the household-level question, among all households, and 28 to 29 (95% CI 22 to 35 for both estimates) percentage points more open defecation among households that received assistance to construct their latrines.” We prefer to focus on the difference in reported open defecation in the abstract because that is the main focus of this paper, rather than raw open defecation rates.

3. The presentation of findings could be improved, the authors tended to explain each table and figure presented in the results rather than state what the findings are.

Response: Thanks for this comment. We have edited the text in the results section to focus more on the findings, rather than explaining each table and figure. We have kept some of the explanatory text because we believe that this explanation helps readers understand the information presented in the tables and figures.

4. It seems that figure 2 and table 2 both present the same results. I recommend that the authors choose which method best summarizes their finding.

Response: Thank you for bringing this up. Yes, the content in figure 2 is also presented in table 2. Table 2 extends on the content presented in figure 2 and presents a more in depth statistical analysis. Although there is some overlap in content here, we believe that the visual presentation of the findings in figure 2 is important for drawing attention to the most important findings of the study. Moreover, since BMJ Open is an online journal, space is less of a constraint. For these reasons, we prefer to keep both figure 2 and table 2.

VERSION 2 – REVIEW

REVIEWER	Robert Ntozini Zvitambo Institute for Maternal and Child health Research, Zimbabwe
REVIEW RETURNED	14-Aug-2019

GENERAL COMMENTS	The authors addressed my comments adequately. No further comments
---